# The Impact of an Early Lifestyle Intervention on Pregnancy Outcomes in a Cohort of Insulin-Resistant Overweight and Obese Women

**DOI:** 10.3390/nu12051496

**Published:** 2020-05-21

**Authors:** Daniela Menichini, Elisabetta Petrella, Vincenza Dipace, Alessia Di Monte, Isabella Neri, Fabio Facchinetti

**Affiliations:** 1International Doctorate School in Clinical and Experimental Medicine, Department of Biomedical, Metabolic and Neural Sciences, University of Modena and Reggio Emilia, 41125 Modena, Italy; 2Unit of Obstetrics and Gynecology, Mother-Infant Department, University of Modena and Reggio Emilia, 41125 Modena, Italy; petrella.elisabetta@gmail.com (E.P.); isabella.neri@unimore.it (I.N.); fabio.facchinetti@unimore.it (F.F.); 3Department of Medical and Surgical Sciences for Mother, Child and Adult, University of Modena and Reggio Emilia, University Hospital Policlinic of Modena, 41125 Modena, Italy; cinzia.dipace@gmail.com (V.D.); themasterplan08@gmail.com (A.D.M.)

**Keywords:** insulin resistance, obesity, pregnancy, lifestyle intervention

## Abstract

Obese women are more likely to have decreased insulin sensitivity and are at increased risk for many adverse pregnancy outcomes. An early lifestyle intervention (LI) may have the potential to reduce the impact of insulin resistance (IR) on perinatal outcomes. We report post hoc analysis of an open-label randomized control trial that includes IR women with body-mass index ≥25 randomly assigned to a LI with a customized low glycemic index diet or to standard care (SC) involving generic counseling about healthy diet and physical activity. Women were evaluated at 16, 20, 28, and 36 weeks of gestation, at which times perinatal outcomes were collected and analyzed. An oral-glucose-tolerance test (OGTT) showed that women in the LI group had lower plasma glucose levels at 120 min at 16–18 weeks of gestation, and at 60 and 120 min at 24–28 weeks. More importantly, these women had a lower rate of large-for-gestational-age (LGA) infants (*p* = 0.04). Interestingly, the caloric restriction and low-glycemic index diet did not increase the rate of small-for-gestational-age (SGA) babies in the LI group. A lifestyle intervention started early in pregnancy on overweight and obese women had the potential to restore adequate glucose tolerance and mitigate the detrimental role of IR on neonatal outcomes, especially on fetal growth.

## 1. Introduction

Obesity is among the risk factors for decreased insulin sensitivity or increased insulin resistance, defined as the decreased response of a nutrient to a given concentration of insulin at target tissue such as liver, muscle, or adipose tissue [1]. When this altered insulin status is prolonged, individuals are prone to developing metabolic syndrome and associated conditions such as diabetes, hypertension, hyperlipidemia, and cardiovascular disorders [2].

Pregnancy is characterized by significant hormonal and metabolic changes in the mother to ensure adequate fetal nutrition [3], and it could amplify insulin resistance. A pregnancy-related 60% decrease in insulin sensitivity puts overweight and obese women at greater risk for complications such as gestational diabetes mellitus (GDM), preeclampsia, and fetal growth abnormalities. Pregnancy was considered a metabolic-stress test for the lifelong risk of metabolic-syndrome development [1,4].

A hyperglycemia and adverse pregnancy outcome (HAPO) study brought to light a correlation between early-pregnancy fasting glucose levels and the earlier onset of GDM [5]. Given the growing global incidence of this disorder and its related complications, strategies for its prevention, diagnosis, and management were suggested [6,7,8].

Lifestyle intervention, including diet and/or physical-activity (PA) advice, has been implemented in several populations. According to comprehensive meta-analysis, clinical relevance was of minor impact, mainly due to variations in the quality of trials, characteristics of the interventions, and assessed populations and outcome definitions between trials [9]. In two previously reported small trials, a hypocaloric-low-glycemic-index diet associated with increased physical activity significantly reduced gestational weight gain (GWG) and GDM rate in overweight/obese women [10,11]. In this study, we aim to evaluate if those clinical improvements could also benefit perinatal outcomes in the subgroup of women showing insulin resistance. The primary objective of the study was to improve the glucose tolerance at the oral glucose-tolerance test (OGTT) at 16–18 and 24–28 weeks of pregnancy. The secondary objective was to improve maternal and neonatal outcomes by reducing the onset of GDM and the rate of large-for-gestational-age (LGA) infants.

## 2. Materials and Methods

### 2.1. Study Design and Protocol

This report is post hoc analysis of an open-label, randomized controlled study. The study was approved by the Ethics Committee of Modena, Area Vasta Emilia Nord (AVEN) in April 2016, reference number 136/15, and registered at clinical.trial.gov as NCT02766426. Pregnant women were recruited among the National Health System (NHS) clinics of the Modena area, and enrolled at the Mother–Infant Department of the University Hospital of Modena. Written informed consent was signed prior to the beginning of the study. All procedures were in accordance with the ethical standards of the responsible committee on human experimentation (institutional and national) and with the Helsinki Declaration of 1975 as revised in 2008.

Women between the 9th and 12th week of pregnancy with body mass index (BMI) ≥25 kg/m^2^, aged ≥18 years, and carrying a singleton pregnancy were eligible to receive the behavioral program as reported below.

The exclusion criteria were chronic diseases including Type 2 diabetes mellitus (DM) and hypertension, previous incidence of GDM, other medical conditions or dietary supplementations that might affect body weight or inflammatory status such as thyroid diseases, previous bariatric surgery, or smoking habits, and any contraindications to practice physical activity.

Eligible women were randomly assigned either to lifestyle intervention (LI) or to standard care (SC) as reported below. The randomization list was obtained using computer-generated random allocation in a 1:1 ratio. The numbers were sealed in numbered envelopes. After eligibility assessment, the midwife/obstetrician opened the envelopes and proceeded with the enrollment.

The women included in the post hoc analysis showed insulin resistance at the first visit in 9–12 weeks of gestation, had homeostatic-model-assessment (HOMA) index ≥ 2.5 maintained during the later stages of gestation, and were 100% compliant with the study program.

### 2.2. Study Procedures

At enrollment, an accurate obstetric, family, and personal history was collected for the assessment of the inclusion/exclusion criteria by a gynecologist and a dietitian. Weight and height were measured to assess BMI as weight (kg)/height (m). This value was considered the prepregnancy BMI.

The study population received a fasting glucose and insulin evaluation during the 12th week of gestation in order to detect subjects with insulin resistance (IR). As recommended for high-risk women, a 75 g OGTT was performed between the 16th and 18th week of gestation, and repeated between the 24th and 28th week of pregnancy in case it tested negative. A diagnosis of GDM was reached for any glucose value exceeding the normal cut-off according to the American Diabetes Association [12].

If the OGTT was positive, women were referred to a diabetologist who decided concerning eventual insulin treatment. Women with GDM were monitored throughout pregnancy as recommended by the National Institute for Health and Care Excellence guidelines [13], with timing and mode of delivery as recommended by the American College of Obstetricians and Gynecologists [14]. Women were scheduled to have specific follow-up evaluations for adherence to the program at the 16th, 20th, 28th, and 36th weeks of pregnancy.

#### 2.2.1. Lifestyle Intervention

Caloric restriction consisted of a low glycemic index, low saturated fat diet with a total intake of 1700 kcal/day. The prescribed diet was characterized by a wide consumption of vegetables, cereals, legumes, and fish, with olive oil as the main source of fat, and no consumption of alcoholic beverages. The diet consisted of three main meals and three snacks (breakfast, snack, lunch, snack, dinner, and evening snack before bedtime). For each meal or snack, the women had several alternatives, all of which were chosen according to individual preferences. The macronutrient composition of the diet was: 55% carbohydrates (80% complex carbohydrates and 20% simple carbohydrates), 20% protein (50% animal and 50% vegetable) and 25% fat (12% mono-unsaturated, 7% polyunsaturated, and 6% saturated) with moderately low saturated fat levels. The daily recommended caloric intake was divided into small frequent meals to avoid ketoacidosis, which frequently occurs because of prolonged fasting. The intake of carbohydrates was at least 225 g/day to prevent ketosis [15].

Each woman was prescribed a PA program that encouraged them to spend 30 min/day walking at least 4 times/week.

#### 2.2.2. Standard Care

Women received a general nutritional booklet concerning proper food intake without specific caloric restriction, in accordance with Italian guidelines [16]. These women also received the same PA program prescription as that of the LI group.

### 2.3. Outcomes Variables

Glucose excursions after the OGTT were studied to evaluate glucose tolerance and diagnose GDM; area under the curve (AUC) was used as an index of glucose excursion [16].

The onset of gestational hypertensive disorder was diagnosed after 20 weeks if elevated blood pressure (systolic ≥140 or diastolic ≥90 mm Hg) without proteinuria was detected [17].

Infants born large for gestational age (LGA) were those having birth weight (BW) ≥90th percentile, while small for gestational age (SGA) were the infants whose BW was ≤10th percentile.

### 2.4. Statistical Analysis

Stata 13.1 (StataCorp. 2013. College Station, TX, USA) was used to analyze data. Statistical tests were designed to compare the impact of lifestyle intervention on pregnancy outcomes in insulin-resistant overweight and obese women (IR in standard care or lifestyle interventions). After interaction was verified, comparisons between groups were made using a two-sided student *t*-test or 1- or 2-way analysis of variance, followed by the Newman–Keuls post hoc multiple-comparisons test. Multivariate logistic regression was then performed to evaluate determinants of any adverse pregnancy outcomes, adjusting for any confounding variables.

Continuous data are reported as mean ± standard deviation (SD). Categorical data are reported as ratios and percentages. All probability values were 2-tailed, and a probability value of <0.05 was considered statistically significant.

## 3. Results

### 3.1. Baseline Characteristics

Eighty-two women out of 217 (30.2%) showed IR according to a HOMA index >2.5 of 9–12 weeks of gestation and were included in the study. Among them, 46 were allocated to the SC group and 36 to the LI group. The two groups had similar baseline sociodemographic and clinical features as reported in Table 1.

### 3.2. Maternal Outcomes

The average gestational weight gain (GWG) during pregnancy was similar between groups. Analyzing GWG according to the Institute of Medicine (IOM) recommendations [18] by dividing the population according to prepregnancy BMI, the categories of women who gained below, within, and above IOM recommendations were balanced among groups (Table 2).

The number of women who developed gestational hypertension (either pregnancy-induced hypertension or pre-eclampsia) was similar in the two groups, as was the rate of other pregnancy complications, including GDM. The gestational age at delivery was the same between groups, as was the incidence of preterm birth, although with a double prevalence in the SC vs the LI group. The rate of induction of labor and cesarean sections was similar between groups (Table 2).

### 3.3. Oral Glucose-Tolerance Test

After a gestation period of 16–18 weeks, women assigned to the SC group had significantly higher mean glucose values at 120 min after consuming 75 g of glucose. The difference was maintained or increased when the OGTT was conducted at 24–28 weeks. Apart from the baseline levels, glucose values were significantly higher in the SC group than those in the LI group (Table 3). The area under the curve (AUC) of glucose release was higher in the SC group, both at 16–18 weeks (*p* = 0.05) and at 24–28 weeks (*p* = 0.02) (Figure 1).

### 3.4. Neonatal Outcomes

Neonatal outcomes are reported in Table 4. Birth weight was similar between groups, as were the rates of macrosomic and SGA babies, while the rate of LGA was significantly higher in the SC group. Low 5-min Apgar scores, the need for resuscitation, and neonatal intensive-care-unit admission were equally distributed between the groups.

Multivariate logistic regression confirmed that women in the SC group were 12% more likely to have an LGA infant (adjusted odds ratio (OR) 1.12 (1.01–1.26)) after adjusting for confounding variables such as BMI at delivery, gestational weight gain and age >40.

## 4. Discussion

Data showed that, in a cohort of overweight/obese women, IR occurs in about one-third of them at the beginning of pregnancy. This may contribute to the long list of adverse outcomes reported in pregnancies of this population, including increased risk for GDM, gestational hypertensive disorders, obstetric interventions, and LGA babies [19,20].

The lifestyle intervention implemented in this study included a low glycemic index, a low-fat hypocaloric diet, moderate physical activity, and specific timing of meals. These interventions were proven to effectively reduce the diet glycemic index (GI) of pregnant women [11]. GI is a measure repeatedly associated with improved perinatal outcomes [21,22].

In this study, lifestyle intervention reduced the plasma glucose response to OGTT, while baseline values remained unaffected. This seems of paramount importance since maternal IR is associated with glucose flux to the fetus and beta-cell dysfunction. Maternal hyperglycemia leads to fetal hyperglycemia and hyperinsulinemia, which, in turn, are responsible for accelerated fetal growth [23,24]. Accordingly, women receiving LI in our trial gave birth to fewer LGA infants than those allocated to SC, although the rate of GDM was similar between the two groups.

Maternal lifestyle interventions did not significantly affect birth weight, and had limited impact on neonatal outcomes [22,25,26,27]. This could be attributable to the specific characteristics of the LI we implemented, which included caloric restriction and several strict follow-ups during pregnancy to monitor adherence to the program. LI did not increase the rate of SGA, suggesting that the restoration of glucose metabolism mitigates the impact of IR while maintaining adequate placental function.

The main limitation of post hoc analysis was the small sample size due to the restrictive inclusion criteria and that we selected only patients who followed the entire program. However, 100% compliance of the patients in both groups meant that the results were not affected by the loss of follow-up.

## 5. Conclusions

Obese women are at increased risk for several adverse pregnancy outcomes and have increased susceptibility to IR. A lifestyle intervention including diet, exercise, and weight control, started early in pregnancy, has the potential to restore adequate glucose tolerance. While their rate of GDM was similar, women receiving LI had lower total glucose secretion than those in the SC group. This mitigates the detrimental role of insulin resistance on neonatal outcomes, reducing the rate of LGA babies.

## Figures and Tables

**Figure 1 nutrients-12-01496-f001:**
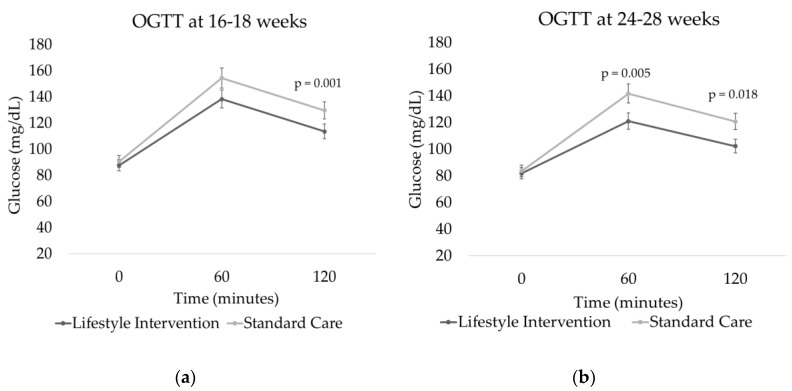
Plasma glucose levels in two groups. Data from OGTT at (**a**) 16–18 and (**b**) 24–28 weeks. Significant differences between the two groups indicated.

**Table 1 nutrients-12-01496-t001:** Maternal socioeconomic and obstetric characteristics.

	Standard Care (*N* = 46)	Lifestyle Intervention (*N* = 36)	*p* Value
**Mean Maternal Age** (y)	30.4 ± 5.5	30.5 ± 4.4	0.98
**Age Class**			0.97
≤25	8 (17.4%)	5 (13.9%)	
26–35	35 (76.1%)	30 (83.4%)	
≥36	3 (6.5%)	1 (2.7%)	
**Education**			0.34
Low	15 (32.6%)	14 (38.9%)	
Medium	26 (56.5%)	15 (41.7%)	
High	5 (10.9%)	7 (19.4%)	
**Occupation**			0.21
Unemployed	20 (43.5%)	13 (36.1%)	
Housewife	12 (26.1%)	8 (22.2%)	
Employed	14 (30.4%)	15 (41.7%)	
**Ethnicity**			0.19
Caucasian	32 (69.8%)	29 (80.5%)	
African	5 (10.8%)	5 (13.9%)	
Sub-Saharan	4 (8.6%)	2 (5.6%)	
Other	5 (10.8%)	0 (0.0 %)	
**Family history of hypertension**	18 (39.1%)	17 (47.2%)	0.46
**Family history of diabetes**	11 (30.5%)	12 (33.3%)	0.34
**Nulliparity**	18 (39.1%)	18 (50.0%)	0.32
**BMI**			0.69
Overweight	7 (15.2%)	5 (13.9%)	
Obese	39 (84.8%)	31 (86.1%)	
**Prepregnancy BMI** (kg/m^2^)	36.7 ± 5.9	37.4 ± 5.5	0.63
**Prepregnancy Weight** (kg)	99.5 ± 15.7	99.0 ± 17.4	0.90

Continuous variables are reported as mean ± standard deviation. Dichotomous variables are reported as numbers and ratios (%).

**Table 2 nutrients-12-01496-t002:** Maternal outcomes according to intervention.

	Standard Care (*N* = 46)	Lifestyle Intervention (*N* = 36)	*p* Value
Gestational Weight Gain (kg)	7.8 ± 7.2	6.7 ± 6.7	0.55
Below Institute of Medicine (IOM)	10 (21.8%)	8 (22.2%)	0.95
Within IOM	18 (39.1%)	17 (47.2%)	0.46
Above IOM	18 (39.1%)	11 (30.5%)	0.42
Gestational Hypertensive Disorders	4 (8.7%)	6 (16.6%)	0.27
Gestational Diabetes Mellitus	22 (47.8%)	17 (47.2%)	0.80
Gestational Age at delivery (w)	38.3 ± 2.01	38.4 ± 1.94	0.76
Preterm Birth	7 (15.5%)	3 (8.3%)	0.37
Induction of Labor	25 (54.3%)	22 (61.1%)	0.53
Caesarean Section	21 (46.7%)	14 (38.9%)	0.89

Continuous variables are reported as mean ± standard deviation. Dichotomous variables are reported as numbers and ratios (%).

**Table 3 nutrients-12-01496-t003:** Glucose profile at 16–18 and 24–28 weeks.

		Standard Care (*N* = 46)	Lifestyle Intervention (*N* = 36)	*p* Value
16–18 weeks	Oral Glucose-Tolerance Test			
	0 min (mg/dL)	90.6 ± 14.7	87.5 ± 10.1	0.35
	60 min (mg/dL)	154.5 ± 39.5	138.4 ± 34.9	0.22
	120 min (mg/dL)	129.6 ± 42.1	113.3 ± 24.1	0.001
	Area-under-curve glucose (mg min/dL)	15,879.1 ± 3891.2	14,340.3 ± 2784.8	0.05
24–28 weeks	Oral Glucose-Tolerance Test			
	0 min (mg/dL)	83.7 ± 7.3	81.8 ± 8.9	0.10
	60 min (mg/dL)	141.8 ± 31.7	120.52 ± 47.8	0.005
	120 min (mg/dL)	120.8 ± 28.1	101.38 ± 39.1	0.018
	Area-under-curve glucose (mg min/dL)	14,645.6 ± 2636.5	12,790.1 ± 2110.2	0.02

Variables reported as mean ± standard deviation. Area under curve (AUC) of glucose levels (mg min/dL) was calculated according to trapezoid rules with those from oral glucose-tolerance test (OGTT).

**Table 4 nutrients-12-01496-t004:** Neonatal outcomes according to intervention.

	Standard Care (*N* = 46)	Lifestyle Intervention (*N* = 36)	*p* Value
Birth Weight (g)	3384.7 ± 648	3343.2 ± 669	0.77
Macrosomia (>4000 g)	7 (15.5%)	3 (8.3%)	0.34
Large for Gestational Age	12 (26.1%)	3 (8.3%)	0.04
Small for Gestational Age	6 (13.3%)	2 (5.5%)	0.25
Apgar at 5 min <7	3 (6.6%)	3 (8.3%)	0.39
pH < 7.1	2 (4.3%)	2 (5.5%)	0.80
Resuscitation	1 (2.2%)	3 (8.3%)	0.21
Neonatal Intensive-Care-Unit Admission	2 (4.4%)	4 (11.1%)	0.26

Continuous variables reported as mean ± standard deviation. Dichotomous variables reported as numbers and ratios (%).

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
