# Peer review of "The Impact of an Early Lifestyle Intervention on Pregnancy Outcomes in a Cohort of Insulin-Resistant Overweight and Obese Women"

_nutrients, 2020, doi:10.3390/nu12051496_

Round 1

Reviewer 1 Report

Although this work have been well performed regarding methodological issues, in my opinion it lacks novelty.

My suggestions are merged in the attached file.

Author Response

Responses to Reviewer 1

We appreciate all the reviewer’s comments. As kindly suggested, we improved the manuscript with the following corrections:

Line 44: we added a comma where suggested.

“Hence, given the growing worldwide incidence of this disorder and its related complications, strategies for its prevention, diagnosis, and management were suggested [6–8].”

Line 47: we changed “clinical impact” with “clinical relevance”

“According to a comprehensive meta-analysis the clinical relevance was of minor impact, mainly due to heterogeneity of interventions [9].”

Lines 118-127: we added a paragraph to describe the main outcomes variables giving details and definitions of the OGTT curve, the AUC, gestational hypertensive disorders and LGA and SGA infants. We also added 2 references, reported below the text.

2.3. Outcomes Variables

The glucose excursion after OGTT was studied to evaluate the glucose tolerance and to diagnose the GDM. Consequently, the glucose area under the curve (AUC) was used as an index of glucose excursion [16].

The onset of gestational hypertensive disorder was diagnosed after 20 weeks if an elevated blood pressure (systolic ≥ 140 or diastolic ≥ 90 mm Hg) without proteinuria, was detected [17].

Infants born large for gestational age (LGA) were those having a birth weight (BW) greater than the 90th percentile for age, while the small for gestational age (SGA) were the infants whose BW was below the 10th percentile for age.

  1. Sakaguchi K, Takeda K, Maeda M, et al. Glucose area under the curve during oral glucose tolerance test as an index of glucose intolerance. Diabetol Int. 2016;7(1):53–58.
  2. Mammaro A, Carrara S, Cavaliere A, et al. Hypertensive Disorders of Pregnancy. J Prenat Med. 2009;3(1):1-5.

Lines 159-160: as recommended by the reviewer we removed the numbers from the text and resumed the results description, since all the numerical details are reported in table 2.

“The gestational age at delivery was the same among groups, as well as the incidence of preterm birth although with a double prevalence in SC vs LI group.”

Lines 170-171: Similarly to the previous correction, we condensed the text and deleted the numerical details from the text because they are reported in table 3 and graphically showed in the figure 1.

“Accordingly, the Area under the Curve (AUC) of glucose release was higher in the SC, both at 16-18weeks (p=0.05) and at 24-28 weeks (p=0.02) (Figure 1).”

Line 173: in the caption of figure #1 we spelled out the acronyms AUC (Area under the curve)

” Variables are reported as Mean ± standard deviation. Area Under the Curve (AUC) of glucose levels (mg min/dl) were calculated according to Trapezoid rules with those from OGTT.”

Table 4: we spelled out NICU as Neonatal Intensive Care Unit

Line 179: As requested by the reviewer, we spelled out SGA and LGA for the first time at lines 125-126.

Lines 207-213: we reported the results of other studies with discordant findings, underling the reason why our intervention may have produced better results in term of reducing the incidence of large for gestational age infants, without increasing the rate of small for gestational age.

“On the contrary, the benefits of maternal lifestyle interventions are disappointing for the offspring. They do not significantly modify birth weight and have a limited impact on neonatal outcomes [22,25-27]. This could be attributable to the specific characteristics of the LI we implemented, which was much “stronger” than others, also including a caloric restriction and several strict follow-ups, scheduled during pregnancy, to monitor the adherence to the program. Nonetheless, it did not increase the rate of SGA, meaning that the restoration of glucose metabolism mitigates the impact of IR still preserving an adequate placental function.”

Reviewer 2 Report

Manuscript “The impact of an early lifestyle intervention on pregnancy outcomes in a cohort of insulin resistant overweight and obese women” (ID: Nutrients-802791).

The objective of the manuscript was to evaluate the impact of an early lifestyle intervention (LI) on pregnancy and perinatal outcomes in the subgroup of overweight and obese women with insulin resistance (IR).

Comments and suggestions for authors:

The manuscript is an interesting article, but requires some consideration. The biggest problem is that the results are limited by significant methodological deficits,
especially for the small sample size. If the summary, conclusions or manuscript are read quickly, it is concluded that the LI started early in pregnancy in IR women reduces the rate of LGA infant. However, it must be borne in mind that the results are very adjusted (SC 12/46 vs LI 3/36, p=0.04) to maintain that statement without qualifying. Although the authors raise these methodological flaws, I think they do so with less emphasis than they deserve. Authors should discuss more about some concepts.

Please find below a list of more specific issues that need to be addressed throughout the manuscript, some related to the line of the manuscript in which they appear:

Line 47. Where says “According to a comprehensive meta-analysis the clinical impact was of minor impact, mainly due to heterogeneity of interventions” (reference 9). However, in said review it is commented that “the data were however strengthened by the relatively low variability between studies”. This meta-analysis (O’Brien et al. 2019) focuses more on the impact of maternal education on response to lifestyle interventions to reduce gestational weight gain. I don't know if it's the right reference. To support your hypothesis perhaps you should have used that of Bain et al. (Diet and exercise interventions for preventing gestational diabetes mellitus. Cochrane Database Syst Rev. 2015 Apr 12;(4):CD010443) or Shepherd et al. (Combined diet and exercise interventions for preventing gestational diabetes. Cochrane Database Syst Rev. 2017 Nov 13;11:CD010443).

Line 50. You provide two references from studies of your group. The 11 says “Obs” instead of “Obstet” in the name of the journal. I cannot locate it in any way.

Line 51. The objective of the study is broad and primary and secondary objectives are not indicated. It would be necessary to calculate the required sample size to verify the proposed objective.

Line 73. As “the women included in the post-hoc analysis were those who resulted insulin resistant at the first visit, between 9-12 weeks of gestation, presenting an homeostatic model assessment (HOMA) index ≥ 2.5”,  and taking into account that insulin resistance increases as the pregnancy progresses, it should be reported promptly when HOMA index was performed in the two study groups.

Line 79. The value of BMI at enrollment, between 9-12 weeks of gestation, was assumed as pre-pregnancy BMI. This may constitute a bias.

Line 123. Eighty-two women were included in the study. Was there no loss during the study? This is inconsistent with the fact that in their previous trial to analyze adherence to a lifestyle program in overweight / obese pregnant women and effect on gestational diabetes mellitus, a quarter of women were lost (reference 10).

Line 126. Table 1. Taking into account that it is more frequent to be overweight than obesity, in their studies (this one and the eference 10) you always include more women with obesity than with overweight. Does this suppose a preferential inclusion of women with obesity over those who are overweight?

Line 154. Table 3. Since very few cases are included in the study, the standard deviations are very wide.

Line 161. In the Materials and Methods section it does not describe what percentile was considered to define SGA and LGA.

Line 183. It is emphasized that “women receiving LI in our trial gave birth to less LGA infants than those allocated to SC”, however LI did not have a substantial effect on other important clinical outcomes.

A logistic regression was not used to evaluate the determinants of the appearance of adverse pregnancy outcomes and the compliance of the early LI with respect to the confounding variables.

Line 187. When proposing intense calorie restriction diets in pregnant women, it is not only necessary to take into account the SGA rate that they could cause, but also other factors such as neonatal hypocalcaemia and others (Mitanchez et al. Nutrients 2020, 12(2), 353).

Line 191. If you selected to present “only the patients who followed the entire program and had complete dataset”, information should be given on those who abandoned the study, as they may be a different population and commit selection bias.

References should be reviewed. For example:

11, 14 and 21: The journal is not “Obs Gynecol” but Obstet Gynecol.

Conclusions. The conclusion of Mitanchez et al. (How Can Maternal Lifestyle Interventions Modify the Effects of Gestational Diabetes in the Neonate and the Offspring? A Systematic Review of Meta-Analyses. Nutrients 2020, 12(2), 353) is more forceful than that expressed by the authors and better summarizes the current state of the matter: “Overall, the benefits of maternal lifestyle interventions are disappointing for the offspring. They do not significantly modify birth weight and have a limited impact on neonatal outcomes...Specific maternal, neonatal and offspring benefits of lifestyle interventions during pregnancy to prevent or improve GDM control or to limit GWG still require clarification”.

Author Response

Responses to Review 2

The manuscript is an interesting article but requires some consideration. The biggest problem is that the results are limited by significant methodological deficits, especially for the small sample size. If the summary, conclusions, or manuscript are read quickly, it is concluded that the LI started early in pregnancy in IR women reduces the rate of LGA infant. However, it must be borne in mind that the results are very adjusted (SC 12/46 vs LI 3/36, p=0.04) to maintain that statement without qualifying. Although the authors raise these methodological flaws, I think they do so with less emphasis than they deserve. Authors should discuss more about some concepts.

Please find below a list of more specific issues that need to be addressed throughout the manuscript, some related to the line of the manuscript in which they appear:

  1. Line 47. Where says “According to a comprehensive meta-analysis the clinical impact was of minor impact, mainly due to heterogeneity of interventions” (reference 9). However, in said review it is commented that “the data were however strengthened by the relatively low variability between studies”. This meta-analysis (O’Brien et al. 2019) focuses more on the impact of maternal education on response to lifestyle interventions to reduce gestational weight gain. I don't know if it's the right reference. To support your hypothesis perhaps you should have used that of Bain et al. (Diet and exercise interventions for preventing gestational diabetes mellitus. Cochrane Database Syst Rev. 2015 Apr 12;(4):CD010443) or Shepherd et al. (Combined diet and exercise interventions for preventing gestational diabetes. Cochrane Database Syst Rev. 2017 Nov 13;11:CD010443).

We appreciate the reviewer’ comment and, as suggested, we referred to the metanalysis of Bain et al. in which results from RCTs of moderate quality suggest no clear difference in the risk of developing GDM for women receiving a combined diet and exercise intervention compared with women receiving no intervention.

Lines 47-49: “According to a comprehensive meta-analysis the clinical relevance was of minor impact, mainly due to the variations in the quality of trials, characteristics of the interventions and populations assessed, and outcome definitions between trials [9]”

  1. Bain E, Crane M, Tieu J, Han S, Crowther C, Middleton P. Diet and exercise interventions for preventing gestational diabetes mellitus. Cochrane Database Syst Rev. 2015;12(4):CD010443.

  1. Line 50. You provide two references from studies of your group. The 11 says “Obs” instead of “Obstet” in the name of the journal. I cannot locate it in any way.

We apologize for the mistake; we corrected the reference as it follows:

  1. Facchinetti F, Vijai V, Petrella E, et al. Food glycemic index changes in overweight/obese pregnant women enrolled in a lifestyle program: a randomized controlled trial. Am J Obstet Gynecol MFM 2019;1:100030.
  2. Line 51. The objective of the study is broad and primary and secondary objectives are not indicated. It would be necessary to calculate the required sample size to verify the proposed objective.

We thank the reviewer for the comment. We specified the primary objective of the study and indicated the secondary objectives at line 52-56.

“Therefore, in this study, we aim to evaluate if those clinical improvements could also benefit perinatal outcomes in the subgroup of women presenting with insulin resistance. More specifically, the primary objective of the study is to improve the glucose tolerance at the OGTT at 16-18 and 24-28 weeks of pregnancy; the secondary objective is to improve maternal and neonatal outcomes reducing the onset of GDM and the rate of Large for Gestational Age (LGA) infants.”

There are discordant opinions on the a posteriori power calculation in a post-hoc analysis because the so-called "post-hoc power analysis" makes little sense in most cases because you have to assume that your estimated effect size is equal to the population effect size, which is very unlikely, unless the sample size is huge. Indeed, our statisticians preferred not to perform the sample size calculation but instead thoroughly selected the subgroups and choosed not to correct for multiple comparisons the results, following the procedure described also by Srinivas et al. and briefly summarized below.

“Correctly performed, post hoc analyses should be directed at discerning patterns or trends by comparing subgroups of the sampled population that were not resolved by the a priori specified endpoint.”

Srinivas T, Ho B, Kang J, Kaplan B. Post Hoc Analyses: After the Facts. Transplantation 2015; 99(1):17–20.

Notably, we choosed not to subject our results to correction for multiple comparisons and offered the results as being exploratory in the context of insulin resistance and the benefits of lifestyle interventions in other populations

  1. Line 73. As “the women included in the post-hoc analysis were those who resulted insulin resistant at the first visit, between 9-12 weeks of gestation, presenting an homeostatic model assessment (HOMA) index ≥ 2.5”,  and taking into account that insulin resistance increases as the pregnancy progresses, it should be reported promptly when HOMA index was performed in the two study groups.

We thank the reviewer for the comment. We clarify that the HOMA index was calculated according to the fasting plasmatic values of glucose and insulin obtained at the first visit between the 9 – 12th weeks of pregnancy. This information is also reported in the text at using the line 86-87:

“The study population received a fasting glucose and insulin evaluation within 12th week of gestation in order to detect subjects with Insulin Resistance (IR).”

  1. Line 79. The value of BMI at enrollment, between 9-12 weeks of gestation, was assumed as pre-pregnancy BMI. This may constitute a bias.

We acknowledge that this assumption may constitute a bias. However, we preferred to use, as the baseline weight, the exact measurements collected by our team at the first visit, considering that the two groups were followed by the same team and women in LI and SC were weighted on the same weight scale. We also referred to a study by Russel et al. that evaluated the accuracy of pregnant women in recalling pre-pregnancy weight and found the under-reporting or underestimating of pre pregnancy weight and over-reporting of GWG.

Russell A, Gillespie S, Satya S, Gaudet LM. Assessing the Accuracy of Pregnant Women in Recalling Pre-Pregnancy Weight and Gestational Weight Gain. JOGC 2013;35(9):802–809.

  1. Line 123. Eighty-two women were included in the study. Was there no loss during the study? This is inconsistent with the fact that in their previous trial to analyze adherence to a lifestyle program in overweight / obese pregnant women and effect on gestational diabetes mellitus, a quarter of women were lost (reference 10).

We appreciate the comment and we apologize for not having clearly reported this information in the inclusion criteria.

We clarify that, being this is a post-hoc analysis, our study population was retrospectively selected, and one of the inclusion criteria was the completeness of the dataset, namely the attendance to all the follow-up and compliance to the behavioral program. Therefore, our sample have been selected to be free of any loss to follow up. We provided to modify the text as it follows:

Lines 77-80:

“The women included in the post-hoc analysis were those who resulted insulin resistant at the first visit, between 9-12 weeks of gestation, presenting an homeostatic model assessment (HOMA) index ≥ 2.5, which attended to all the follow-ups during pregnancy, showing the 100% of the compliance to study program.”

We also specified the lost to follow-up rate in the result section, lines 142-144:

“Ninety-three women out of 217 (42.8%) showed IR according to HOMA index > 2.5 between 9-12 weeks, of them the 11.8% were lost to follow-up (5 women in the SC and 6 in the LI group) and therefore excluded from the study. Overall, a total of 82 IR women were included in the study.”

  1. Line 126. Table 1. Taking into account that it is more frequent to be overweight than obesity, in their studies (this one and the reference 10) you always include more women with obesity than with overweight. Does this suppose a preferential inclusion of women with obesity over those who are overweight?

We thank the reviewer for the question. The higher rate of obese women reported in our study population is because we included insulin resistant women, presenting an HOMA ≥ 2.5 at the first visit (9-12th weeks). Therefore, it is more likely that an obese woman is insulin resistant, than an overweight one.

Kahn S, Hull R, Utzschneider K. Mechanisms linking obesity to insulin resistance and type 2 diabetes. Nature 2006; 444, 840–846

.

  1. Line 154. Table 3. Since very few cases are included in the study, the standard deviations are very wide.

We acknowledge the small sample size as the major limitation of our study. (Lines 206-207)

  1. Line 161. In the Materials and Methods section, it does not describe what percentile was considered to define SGA and LGA.

We added in the M&M section a paragraph explaining in details the main outcomes variables and we described thoroughly the percentiles to which we referred.

Lines 125-127:

“Infants born large for gestational age (LGA) were those having a birth weight (BW) greater than the 90th percentile for age, while the small for gestational age (SGA) were the infants whose BW was below the 10th percentile for age.”

  1. Line 183. It is emphasized that “women receiving LI in our trial gave birth to less LGA infants than those allocated to SC”, however LI did not have a substantial effect on other important clinical outcomes.

The reduction in LGA infants was one of the main expected outcomes, considering that women with an altered glucose metabolism are more likely to deliver a LGA or macrosomic baby. However, we specified that the GDM rate was similar between groups.

Lines 205-206:

“Accordingly, women receiving LI in our trial gave birth to less LGA infants than those allocated to SC, although the rate of GDM was similar between the two groups.”

  1. A logistic regression was not used to evaluate the determinants of the appearance of adverse pregnancy outcomes and the compliance of the early LI with respect to the confounding variables. 

We thank the reviewer for the comment. As suggested, we performed the logistic regression to evaluate the determinants of LGA in our study, thus we adjusted for confounding variables such as BMI at delivery, gestational weight gain and age > 40. And we found that likelihood of having a large for gestational age infant is 12 times higher in the SC group. Adjusted OR 1.12 (1.01 - 1.26).

We reported this datum at line 182-184

“The multivariate logistic regression confirmed that women in the SC group were 12% more likely to have a LGA infant [adjusted OR 1.12 (1.01 - 1.26)], adjusting for confounding variables such as BMI at delivery, gestational weight gain and age > 40.”

We also added the multivariate logistic regression test into the statistical analysis paragraph in the M&M section as it follows:

Lines 135-137

“A multivariate logistic regression was then performed to evaluate the determinants of the appearance of adverse pregnancy outcomes and the compliance of the early LI adjusting for confounding variables.”

  1. Line 187. When proposing intense calorie restriction diets in pregnant women, it is not only necessary to take into account the SGA rate that they could cause, but also other factors such as neonatal hypocalcaemia and others (Mitanchez et al. Nutrients 2020, 12(2), 353).

We appreciate the interesting observation. Unfortunately, we are not able to collect this information retrospectively, but we will for sure add this variable in the future studies on LI in pregnancy.

So far, to evaluate the newborn health status, we were able to report the Apgar score, the pH values, and the need for resuscitation or NICU admission.

  1. Line 191. If you selected to present “only the patients who followed the entire program and had complete dataset”, information should be given on those who abandoned the study, as they may be a different population and commit selection bias.

We appreciate the observation. The primary RCT study from which we selected our sub-groups for this post-hoc analysis, is the study by Facchinetti et al. (reference number 11) as reported below:

  1. Facchinetti F, Vijai V, Petrella E, et al. Food glycemic index changes in overweight/obese pregnant women enrolled in a lifestyle program: a randomized controlled trial. Am J Obstet Gynecol MFM 2019;1:100030.

To evaluate the baseline characteristics of women lost to follow-up, we selected from the original database, the insulin resistant women at the first visit who were lost to follow-up, thus were excluded from the final analysis.

The lost to follow-up rate of our post-hoc analysis was 10% in the SC group and 16% in the LI intervention group. Therefore, 5 women in the SC group and 6 women in the LI group were excluded from the analysis. Baseline characteristics were similar to those of women remaining in the study, following all the scheduled appointments.

We added this information to the manuscript at lines 142-146

“Ninety-three women out of 217 (42.8%) showed IR according to HOMA index > 2.5 between 9-12 weeks, of them the 11.8% were lost to follow-up (5 women in the SC and 6 in the LI group) and therefore excluded from the study. Overall, a total of 82 IR women were included in the study. Among them, 46 were allocated to the SC group and 36 women in the LI group. The two groups had similar baseline sociodemographic and clinical features as reported in Table 1.”

  1. References should be reviewed. For example: 11, 14 and 21: The journal is not “Obs Gynecol” but Obstet Gynecol.

We thank the reviewer for the observation. We corrected the wrong citations.

  1. Facchinetti F, Vijai V, Petrella E, et al. Food glycemic index changes in overweight/obese pregnant women enrolled in a lifestyle program: a randomized controlled trial. Am J Obstet Gynecol MFM 2019;1:100030.

  1. American College of Obstetricians and Gynecologists. ACOG Practice Bulletin No. 190: gestational diabetes mellitus. Obstet Gynecol 2018; 131: e49–64.

  1. Brand-Miller A, Hayne S, Petocz P, Colagiuri S. Low-glycemic index diets in the management of diabetes: a meta-analysis of randomized controlled trials. Diabetes Care 2003; 26:2261–7.

  1. The conclusion of Mitanchez et al. (How Can Maternal Lifestyle Interventions Modify the Effects of Gestational Diabetes in the Neonate and the Offspring? A Systematic Review of Meta-Analyses. Nutrients 2020, 12(2), 353) is more forceful than that expressed by the authors and better summarizes the current state of the matter: “Overall, the benefits of maternal lifestyle interventions are disappointing for the offspring. They do not significantly modify birth weight and have a limited impact on neonatal outcomes...Specific maternal, neonatal and offspring benefits of lifestyle interventions during pregnancy to prevent or improve GDM control or to limit GWG still require clarification”.

We appreciate the suggestion and thus we modified the text including the systematic review of Mitanchez et al. among the studies that reported findings discordant to ours.

Lines 207-209:

“On the contrary, the benefits of maternal lifestyle interventions are disappointing for the offspring. They do not significantly modify birth weight and have a limited impact on neonatal outcomes [22,25-27].”

Round 2

Reviewer 2 Report

The authors reply to the observations presented and make corrections to the manuscript that I believe clarify it. However, there is a pending question that I would like you to comment on.

In this update you describe that “Infants born large for gestational age (LGA) were those having a birth weight (BW) greater than the 90th percentile for age, while the small for gestational age (SGA) were the infants whose BW was below the 10th percentile for age”, and also that “the primary RCT study from which we selected our sub-groups for this post-hoc analysis, is the study by Facchinetti et al. (reference number 11: Facchinetti F, Vijai V, Petrella E, et al. Food glycemic index changes in overweight/obese pregnant women enrolled in a lifestyle program: a randomized controlled trial. Am J Obstet Gynecol MFM 2019;1:100030). I have finally located this study, which is not in Pubmed, and I do not find in that manuscript the percentile classification for LGA and SGA. However, in the other reference presented with the same study population (reference number 10. Bruno R, Petrella E, Bertarini V, Pedrielli G, Neri I, Facchinetti F. Adherence to a lifestyle programme in overweight/obese pregnant women and effect on gestational diabetes mellitus: a randomized controlled trial. Matern Child Nutr 2017; 13:1–11) clear reference is made to the delimitation of the percentiles to consider SGA and LGA (Research design and methods / Data collection / “Large for gestational age (LGA) was defined as a birth weight ≥ the 90th centile; small for gestational age (SGA) was defined as a birth weight ≤ the 10th centile”), which does not coincide with what you now present. Could you tell me if this is a mistake and if this consideration could alter the results presented in this current study or in the previous study cited?

In the previous reference 21 (current 23. Catalano et al.) The name of the journal "Obs" has not been corrected by "Obstet" as I previously suggested.

Author Response

The authors reply to the observations presented and make corrections to the manuscript that I believe clarify it. However, there is a pending question that I would like you to comment on.

In this update you describe that “Infants born large for gestational age (LGA) were those having a birth weight (BW) greater than the 90th percentile for age, while the small for gestational age (SGA) were the infants whose BW was below the 10th percentile for age”, and also that “the primary RCT study from which we selected our sub-groups for this post-hoc analysis, is the study by Facchinetti et al. (reference number 11: Facchinetti F, Vijai V, Petrella E, et al. Food glycemic index changes in overweight/obese pregnant women enrolled in a lifestyle program: a randomized controlled trial. Am J Obstet Gynecol MFM 2019;1:100030). I have finally located this study, which is not in Pubmed, and I do not find in that manuscript the percentile classification for LGA and SGA. However, in the other reference presented with the same study population (reference number 10. Bruno R, Petrella E, Bertarini V, Pedrielli G, Neri I, Facchinetti F. Adherence to a lifestyle programme in overweight/obese pregnant women and effect on gestational diabetes mellitus: a randomized controlled trial. Matern Child Nutr 2017; 13:1–11) clear reference is made to the delimitation of the percentiles to consider SGA and LGA (Research design and methods / Data collection / “Large for gestational age (LGA) was defined as a birth weight ≥ the 90th centile; small for gestational age (SGA) was defined as a birth weight ≤ the 10th centile”), which does not coincide with what you now present. Could you tell me if this is a mistake and if this consideration could alter the results presented in this current study or in the previous study cited?

We really appreciate the comment. We apologize for the mistake.

We clarify that, being this post-hoc analysis obtained by the same study population of the two studies the reviewer cited (Facchinetti et al. 2019 and Bruno et al. 2017), the delimitation of the percentiles to consider SGA and LGA was the same and we ensure that this mistake in reporting the definition does not affect or alter the results presented.

We provided the correct definition in the text at lines 125-126:

 Infants born large for gestational age (LGA) were those having a birth weight (BW) ≥ 90th centile, while the small for gestational age (SGA) were the infants whose BW was ≤ 10th centile.

In the previous reference 21 (current 23. Catalano et al.) The name of the journal "Obs" has not been corrected by "Obstet" as I previously suggested.

We apologize for the oversight. Please find below and in the text the corrected reference:

  1. Catalano P, Huston L, Amini S, Kalhan S. Longitudinal changes in glucose metabolism during pregnancy in obese women with normal glucose tolerance and gestational diabetes mellitus. Am J Obstet Gynecol 1999; 180:903–916.
